# UQ for Credit Risk Management: A deep evidence regression approach

## Abstract

Machine Learning has invariantly found its way into various Credit Risk applications. Due to the intrinsic nature of Credit Risk, quantifying the uncertainty of the predicted risk metrics is essential, and applying uncertainty-aware deep learning models to credit risk settings can be very helpful. In this work, we have explored the application of a scalable UQ-aware deep learning technique, Deep Evidence Regression and applied it to predicting Loss Given Default. We contribute to the literature by extending the Deep Evidence Regression methodology to learning target variables generated by a Weibull process and provide the relevant learning framework. We demonstrate the application of our approach to both simulated and real-world peer to peer lending data.

## 1 Introduction

### 1.1 Credit Risk Management

Credit risk management is assessing and managing the potential losses that may arise from the failure of borrowers or counterparties to fulfil their financial obligations. In other words, it identifies, measures, and mitigates the risks associated with lending money or extending credit to individuals, businesses, or other organizations.

Credit risk's anticipated loss (EL) comprises three components: Probability of Default (PD), Loss Given Default (LGD), and Exposure at Default (EAD). PD is the likelihood that a borrower will fail to fulfill their financial commitments in the future. LGD refers to the proportion of the outstanding amount that is lost in the event of default. Lastly, EAD refers to the outstanding amount at the time of default.[8]

LGD prediction is important as accurate prediction of LGD not only supports a healthier and risk-less allocation of capital, but is also vital for pricing the security properly. [8] & [14]. There is a large body of literature using advanced statistical and machine learning methods for prediction of LGD [8]. However the machine learning literature on LGD has yet to address an essential aspect, which is the uncertainty surrounding the estimates and predictions.[4].

UQ techniques like Bayesian Neural Network, Monte Carlo Dropout and ensemble methods as outlined in [1] present a natural first step towards quantifying uncertainty. However, almost all these methods are computationally and memory intensive, and require sampling on test data after fitting the network, making them difficult to adapt for complex neural network architectures that involve a large number of parameters.

## 1.2 Deep Evidence Regression

The primary inspiration of this work is taken from the work done by Amini et al in [2]. The paper develops a unique approach, **Deep Evidence Regression** as a scalable and accurate UQ aware deep learning technique for regression problems. This approach predicts the types of uncertainty directly within the neural network structure, by learning prior distributions over the parameters of the target distribution, referred to as evidential distributions. Thus this method is able to quantify uncertainty without extra computations after training, since the estimated parameters of the evidential distribution can be plugged into analytical formulas for epistemic and aleatoric uncertainty, and target predictions.

The setup of the problem is to assume that the observations from the target variable, $y_i$ are drawn i.i.d. from a **Normal distribution** with unknown mean and variance parameters $\theta = \mu, \sigma^2$. With this we can write the log likelihood of the observation as:

$$Lik(\mu, \sigma^2) = log(p(y_i|\mu, \sigma^2) = -\frac{1}{2}\log(2\pi\sigma^2) - \frac{(y_i - \mu)^2}{2\sigma^2}$$

Learning $\theta$ that maximises the above likelihood successfully models the uncertainty in the data, also known as the aleatoric uncertainty. However, this model is oblivious to its predictive epistemic uncertainty. [2]. Epistemic uncertainty, is incorporated by placing higher-order prior distributions over the parameters $\theta$. In particular a Gaussian prior is placed on the unknown mean and an Inverse-Gamma prior on the unknown variance.

$$\mu \sim \mathcal{N}(\gamma, \sigma^2\nu^{-1}) \quad \sigma^2 \sim \Gamma^{-1}(\alpha, \beta)$$

Following from above the posterior $p(\mu, \sigma^2|\gamma, \nu, \alpha, \beta)$ can be approximated as $p(\mu|\gamma, \nu) * p(\sigma^2|\alpha, \beta)$. Hence:

$$p(\mu, \sigma^2|\gamma, \nu, \alpha, \beta) = \frac{\beta^\alpha \sqrt{\nu}}{\Gamma(\alpha)\sqrt{2\pi\sigma^2}}(1/\sigma^2)^{\alpha+1} \exp\left(-\frac{2\beta + \nu(\gamma - \mu)^2}{2\sigma^2}\right)$$

Amini et al [2] thus find the likelihood of target variable given evidential parameters, as:

$$p(y_i|\gamma, \nu, \alpha, \beta) = \int_\theta p(y_i|\theta)p(\theta|\gamma, \nu, \alpha, \beta)$$

where $\theta = \{\mu, \sigma^2\}$. Then a Neural Network is trained t infer, the parameters $m = \{\gamma, \nu, \alpha, \beta\}$, of this higher-order, evidential distribution.

## 1.3 Weibull distribution

The Weibull distribution is a continuous probability distribution commonly used in reliability analysis to model the failure time of a system or component.[7] The probability density function (PDF) of the Weibull distribution is given by:

$$f(x; \lambda, k) = \begin{cases} \frac{k}{\lambda}\left(\frac{x}{\lambda}\right)^{k-1} e^{-(x/\lambda)^k} & \text{if } x \geq 0, \\ 0 & \text{if } x < 0, \end{cases} \tag{1}$$

where $\lambda > 0$ is the scale parameter and $k > 0$ is the shape parameter. The scale parameter determines the location of the distribution, while the shape parameter controls the rate at which the failure rate changes over time. There is a body of lietrature that explores the application of the weibull distribution to various credit risk applications. [9] [11].

The work by [12] assumes a normal distribution on LGD values. While this assumption might be true in a lot of settings, however it does not follow in the context of Loss Given Default. While normal distribution is symmetric and has a support over entire real line, however the LGD values are restricted to a range of $[0, 1]$ and might not necessarily be symmetric.

Hence in the section below we provide a novel theoretical framework to learn target variables which follow Weibull distribution. We provide the following theoretical results, in the setting of target variables following a Weibull dataset.

- Log Likelihood

- Mean Prediction

- Prediction Uncertainty

We also provide results testing our approach on both simulated and real world dataset.

## 2 Deep Evidence Regression for Weibull Data

### 2.1 Problem setup

We consider the problem where the observed targets, $y_i$, are drawn iid from a Weibull distribution, with a known shape or rate parameter k and an unknown scale $\lambda$. Although ideally we would want to keep both the parameters unknown, however with both $\lambda$ and $k$ there are no priors with which likelihood can be computed analytically [3]. Hence we have decided to simplify the problem setup by assuming known shape $k$.

$$y_i \sim Weibull(k, \lambda) \tag{2}$$

$$\text{where } k \in \mathrm{R}^+, \lambda \in \mathrm{R}^+ \tag{3}$$

$$\text{Hence pdf of } y_i \text{ is} \tag{4}$$

$$\implies p(y_i; \lambda, k) = \begin{cases} \frac{k}{\lambda}\left(\frac{y_i}{\lambda}\right)^{k-1} e^{-(y_i/\lambda)^k}, & \text{if } y_i \geq 0 \\ 0, & \text{otherwise} \end{cases} \tag{5}$$

For the above setting we want to place priors on the unknown parameter, $\lambda$, such that we are able to get solve for the likelihood of $y_i$ given the parameters of the prior distribution. Hence similar to work in [17] and [5], we define the following prior.

$$\theta = \lambda^k \tag{6}$$

$$\text{Hence the pdf of } y_i \text{ becomes:} \tag{7}$$

$$p(y_i|\theta, k) = \frac{k}{\theta} y_i^{k-1} \exp\left(-y_i^k/\theta\right) \tag{8}$$

$$\text{And we place a Inverse Gamma Prior on } \theta \tag{9}$$

$$\theta \sim \Gamma(\alpha, \beta) \quad (\alpha > 2) \tag{10}$$

$$\text{Hence pdf of } \theta \text{ is} \tag{11}$$

$$p(\theta|\alpha, \beta) = \frac{\beta^\alpha}{\Gamma(\alpha)} \frac{1}{\theta^{\alpha+1}} \exp\left(-\frac{\beta}{\theta}\right) \tag{12}$$

### 2.2 Learning Log-Likelihood

Hence we can define likelihood of $y_i$ given the higher order evidential parameters $\alpha, \beta$ can be defined as :

$$Lik = p(y_i|\alpha, \beta) = \int_{\theta} p(y_i|\theta, k)p(\theta|\alpha, \beta)d\theta \tag{13}$$

$$\text{Now given } \lambda, k > 0 \implies \theta > 0 \tag{14}$$

$$p(y_i|\alpha, \beta) = \int_{\theta=0}^{\infty} \left(\frac{k}{\theta}y_i^{k-1}\exp\left(-y_i^k/\theta\right)\right)\left(\frac{\beta^{\alpha}}{\Gamma(\alpha)}\frac{1}{\theta^{\alpha+1}}\exp\left(-\frac{\beta}{\theta}\right)\right)d\theta \tag{15}$$

$$= ky_i^{k-1}\frac{\beta^{\alpha}}{\Gamma(\alpha)}\int_{\theta=0}^{\infty}\frac{1}{\theta^{\alpha+2}}\exp\left(-\frac{y_i^k+\beta}{\theta}\right) \tag{16}$$

$$= ky_i^{k-1}\frac{\beta^{\alpha}}{\Gamma(\alpha)}\frac{\Gamma(1+\alpha)}{(y_i^k+\beta)^{1+\alpha}} \tag{17}$$

$$= \frac{\alpha k y_i^{k-1}\beta^{\alpha}}{(y_i^k+\beta)^{\alpha+1}} \tag{18}$$

Hence the log-likelihood for i'th observation is defined as:

$$Log - Lik_i = L_i^{lik} = \log\alpha_i + \log k + (k-1)\log y_i + \alpha_i\log\beta_i - (\alpha_i + 1)(y_i^k + \beta_i) \tag{19}$$

We set up our neural network to minimise the negative Log-Likelihood plus some regularisation cost, discussed in section below.

## 2.3 Mean Prediction and UQ of prediction

Given the main advanatge of Deep Evidence Regression over other UQ aware deep learning methods like Bayesian NN, esembling etc, is due to existence of analytical solution for both predictions and unceratinty from NN output, without the need for sampling. Hence this section details the derivation of mean prediction and total prediction uncertainty.

### 2.3.1 Mean Prediction

$$\text{We define the mean prediction as } E[Z|\alpha, \beta] \tag{20}$$

$$\text{where } Z = E[y_i] \tag{21}$$

$$\text{Now given } y_i \sim Weibull(k, \lambda) \tag{22}$$

$$E[Z] = E[\lambda * \Gamma(1+\frac{1}{k})] = E(\lambda) * \Gamma(1+\frac{1}{k}) \quad \text{(k is known)} \tag{23}$$

$$E[\lambda] = \int_{\lambda}\lambda p(\lambda)d\lambda \tag{24}$$

$$\tag{25}$$

Hence to solve for mean prediction we need to find pdf $p(\lambda)$. Because we know $\theta = \lambda^k \sim \Gamma^{-1}(\alpha, \beta)$, we can use change of variable to find pdf of $\lambda$ [18].

$$\implies E[\lambda|\alpha, \beta] = \frac{k\beta^{\alpha}}{\Gamma(\alpha)} * \Gamma(\frac{k\alpha-1}{k}) * \frac{1}{k} * \frac{1}{\beta^{\frac{k\alpha-1}{k}}} \tag{26}$$

The mean prediction can thus be simplified as:

$$E[Z|\alpha, \beta] = E(\lambda) * \Gamma(1+\frac{1}{k}) \tag{27}$$

$$= \frac{k\beta^{\alpha}}{\Gamma(\alpha)} * \Gamma(\frac{k\alpha-1}{k}) * \frac{1}{k} * \frac{1}{\beta^{\frac{k\alpha-1}{k}}} * \Gamma(1+\frac{1}{k}) \tag{28}$$

$$= \Gamma(1+\frac{1}{k})\frac{1}{\Gamma(\alpha)}\Gamma(\alpha-\frac{1}{k}) * \beta^{1/k} \tag{29}$$

### 2.3.2 Prediction Uncertainty

We quantify the total uncertainty as $Var(Z)$ with defined as above, i.e. $Z = E[y_i]$

$$Var(Z|\alpha, \beta) = Var(\lambda) * \Gamma^2(1 + \frac{1}{k}) \tag{30}$$

$$= (E[\lambda^2] - E[\lambda]) * \Gamma^2(1 + \frac{1}{k}) \tag{31}$$

$$\text{With } E(\lambda) \text{ defined as in 26, we only need } E(\lambda^2) \tag{32}$$

$$E[\lambda^2] = \int_\lambda \lambda^2 p(\lambda) d\lambda \tag{33}$$

$$\tag{34}$$

Similar to approach outlined in 2.3.1, we get:

$$E[\lambda^2|\alpha, \beta] = \Gamma(\frac{k\alpha - 2}{k})\frac{\beta^{2/k}}{\Gamma(\alpha)} \tag{35}$$

Hence we can write

$$Var(Z) = \Gamma^2(1 + \frac{1}{k}) * \left[\Gamma(\frac{k\alpha - 2}{k})\frac{\beta^{2/k}}{\Gamma(\alpha)} - \left(\Gamma(\frac{k\alpha - 1}{k})\frac{\beta^{1/k}}{\Gamma(\alpha)}\right)^2\right] \tag{36}$$

or

$$Var(Z) \propto \frac{\beta^{2/k}}{\Gamma(\alpha)^2}\left[\Gamma(\alpha)\Gamma(\frac{k\alpha - 2}{k}) - \Gamma^2(\frac{k\alpha - 1}{k})\right]$$

## 2.4 Regularisation Cost

In this section, we outline the process of regularization during training by implementing a regularisation penalty, which involves assigning a high uncertainty cost. The purpose of this penalty is to inflate the uncertainty associated with incorrect predictions, thereby improving the overall efficacy of the model. As followed in [2], the intuition behind the regularisation cost is to increases the variance of prediction in cases where it's unsure. This utility of this approach has been demonstrated in classification setting by [13] and in regression setting by [2].

Hence we define the Regularisation cost for the i'th observation as

$$L_i^{reg} = |error_i| * (\frac{\alpha_i}{\beta_i})$$

where $error_i = y_i - Z_i$ and Z is defined in 27.

**Note** The regularization cost mentioned earlier has been determined as the most effective through experimentation. However, in order to precisely determine the coefficients of $\alpha$ and $\beta$ in the regularization cost, further theoretical analysis is required. By conducting a deeper theoretical investigation, we can establish the optimal values for these coefficients, which will enhance the regularization process and improve the overall performance of the model.

## 2.5 Neural Network training schematic

The learning process is then set up with a deep neural network with two output neurons to predict the parameters of the prior/evidential distribution, $\alpha$ and $\beta$. The neural network is trained using the cost function:

$$L_i^{NN} = -L_i^{lik} + c * L_i^{reg}$$

where c is the hyper-parameter governing the strength of regularisation.

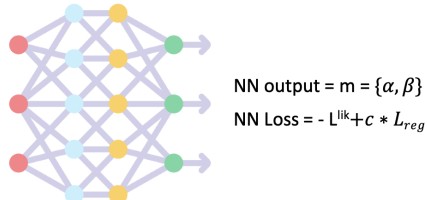

Figure 1: NN training schematic

## 3 Results and Experiments

In this section, we present the results of experiments conducted on both simulated and real data. The aim was to evaluate the performance of our proposed method and compare it with existing methods. The simulated data was generated based on example data given in [2], while the real dataset was obtained from a peer to peer lending company.

### 3.1 Simulated Data

Here generate a target variable following a Weibull distribution. The target variable is generated as:

$$y_i = x_i^2 + \epsilon, \epsilon \sim Weibull(k = 1.6, \lambda)$$

The train set is comprised of uniformly spaced $x \in [-4, 4]$ while test set is $x \in [-5, 5]$. The value of $\lambda$ is varied between $[0.2, 0.4]$ to test the effect of noise magnitude on the approach.

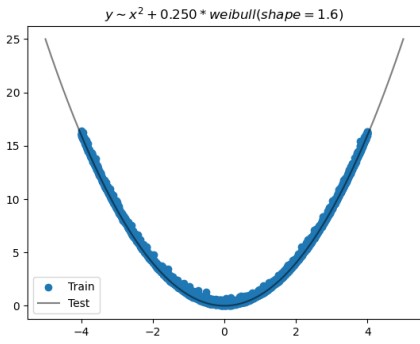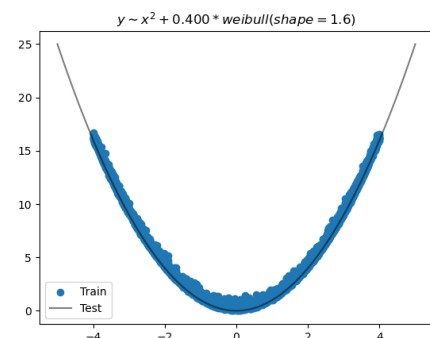

Figure 2: Synthetic data generated for varying $\lambda$

Next we fit both the original deep evidence regression and proposed weibull version of deep evidence regression. Since our approach assumes known $k$, k is estimated from the training set.

Comparing the two versions qualitatively, we observe that the original model's predictions exhibit consistent uncertainty regardless of whether the data is within or outside the distribution. In contrast, the proposed version demonstrates improved capability in capturing prediction uncertainty. The proposed model's prediction interval gradually expands beyond the training data range $|x| > 4$, indicating its ability to account for uncertainty in Out-of-distribution data. On comparison, for the benchmark version of the model displays a slightly narrower prediction interval at the edges of the training window, contrary to expectations of interval widening.

Quantitatively to assess the performance of our proposed method, we compare it to the benchmark model by evaluating key metrics such as mean squared error (MSE) and negative log-likelihood (NLL).

We can see that the proposed version exhibits significantly lower negative log likelihood compared to the original Deep Evidence Regression model. This indicates that the proposed model better

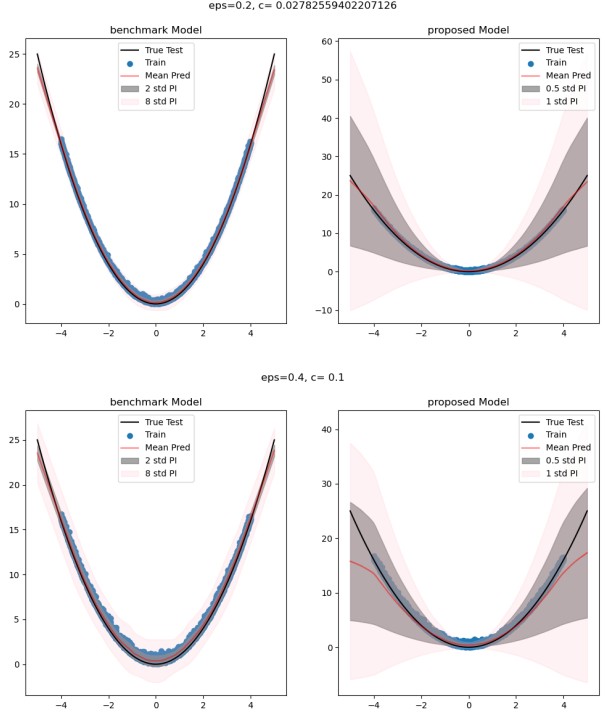

Figure 3: Deep evidence regression (left) vs Weibull evidence Regression (right). We see that uncertainty is much better captured by proposed version.

Table 1: Deep Evidence regression original vs proposed results on simulated data for varying $\lambda$. We see MSE (or mean squared error) is similar for both, while NLL (or Negative log likelihood) values are much better captured by proposed version.

| $\lambda$ | MSE(test) | | NLL(test) | |
|---|---|---|---|---|
| | benchmark | proposed | benchmark | proposed |
| 0.2 | **0.099303** | **0.571519** | 70.64365 | **7.416122** |
| 0.25 | **0.119299** | **0.504875** | 41.71667 | **6.667958** |
| 0.3 | **0.142722** | 3.871369 | 36.16713 | **6.202275** |
| 0.35 | **0.143117** | 3.202328 | 57.02918 | **5.773156** |
| 0.4 | **0.172697** | 8.981477 | 42.53032 | **5.471559** |

aligns with the actual distribution of the data, capturing the uncertainty more accurately. However, despite this improvement, the original model outperforms in capturing the underlying signal beyond the training window, as evidenced by its lower mean squared error (MSE) values.

## 3.2 Real Data: Loss Given Default for peer to peer lending

In this subsection, we showcase the utility of our proposed learning approach by extending it to the intricate and complex domain of credit risk management in the context of peer to peer lending. Peer-to-peer lending, which is an emerging form of credit aimed at funding borrowers from small lenders and individuals seeking to earn interest on their investments. Through an online platform, borrowers can apply for personal loans, which are typically unsecured and funded by one or more peer investors. The P2P lender acts as a facilitator of the lending process and provides the platform, rather than acting as an actual lender.

Credit risk management is crucial for peer-to-peer lending data as it helps mitigate the potential default risks associated with borrowers, ensuring a healthier loan portfolio and reducing financial

loses. By effectively analyzing and managing credit risk, P2P lending platforms can maintain investor confidence, attract more participants, and sustain the long-term viability of the lending ecosystem.

The dataset under consideration pertains to peer to peer mortgage lending data during the period of 2007 to 2014 sourced from Kaggle [16]. However, the data does not include the loss given default values. Instead, the recovery rate has been used as a proxy, which is calculated as the ratio of recoveries made to the origination amount. The dataset contains approximately 46 variables denoted as 'x,' which include features such as the time since the loan was issued, debt-to-income ratio (DTI), joint applicant status, and delinquency status, among others. In total, the dataset comprises around 23,000 rows.

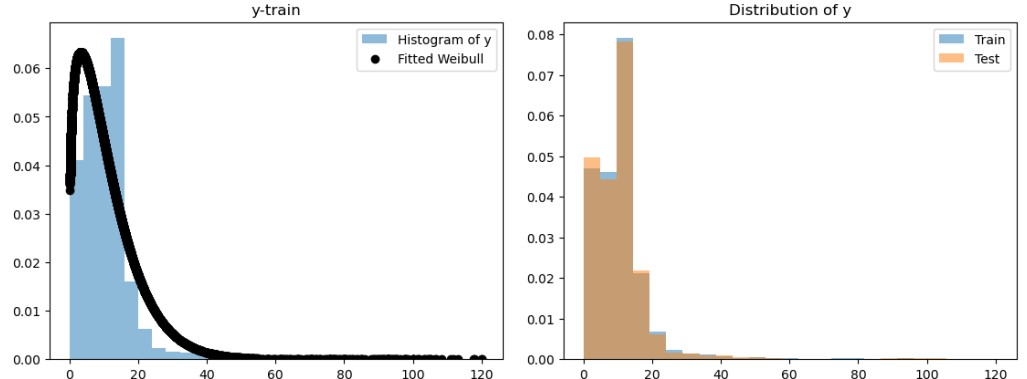

Figure 4: Distribution and Weibull fit of the recovery rate (left). Histogram of recovery rate for train/test split (right). It appears that Weibull distribution might not be a good fit to this data.

As described in the approach the shape parameter was found as $k = 1.254$ by fitting a Weibull distribution on the train dataset. Also given the sensitivity of both the approaches to regression cost, hyper parameter optimisation was done to arrive at the best regularisation cost. After arriving at the best regularisation cost 10 trials of neural network training were conducted with this best regularization cost for benchmark and proposed model separately.

Similar to the synthetic case, we see that the proposed model demonstrates superior performance compared to the benchmark model in terms of mean squared error (MSE) and negative log likelihood. Additionally, it exhibits the ability to generate more accurate prediction intervals. The difference in performance between the benchmark and proposed models is even more pronounced in this case compared to the simulated data, and the benchmark fails even to retrieve the underlying signal, let alone the prediction uncertainty. To reinforce this, the benchmark model was also run with 0 regularisation cost and it was found to not improve the MSE. This behaviour outlines the difficulty of tuning regularisation parameter for benchmark model. It is also worth noting that the benchmark model also predicts $\infty$ as uncertainty for a significant number of observations.

Table 2: Results for original vs proposed model for recovery rate. proposed version does not only has both lower MSE and NLL

|  | MSE | | NLL | |
|---|---|---|---|---|
|  | benchmark | proposed | benchmark | proposed |
| *test* | $84.333 \pm 0.352$ | $\mathbf{40.498 \pm 4.745}$ | $2.757 \pm 0.012$ | $\mathbf{2.311 \pm 0.024}$ |
| *train* | $84.320 \pm 0.216$ | $\mathbf{39.909 \pm 4.911}$ | $2.766 \pm 0.009$ | $\mathbf{2.314 \pm 0.0228}$ |

## 4 Related Work

This work is primarily inspired by the work done by Amini et al [2], which proposes Deep Evidentail Learning approach. Maximillian et al [12] have used the same approach and shown it's utility to Loss given default for bonds from Moody's recovery database. Additionally there's a huge body of

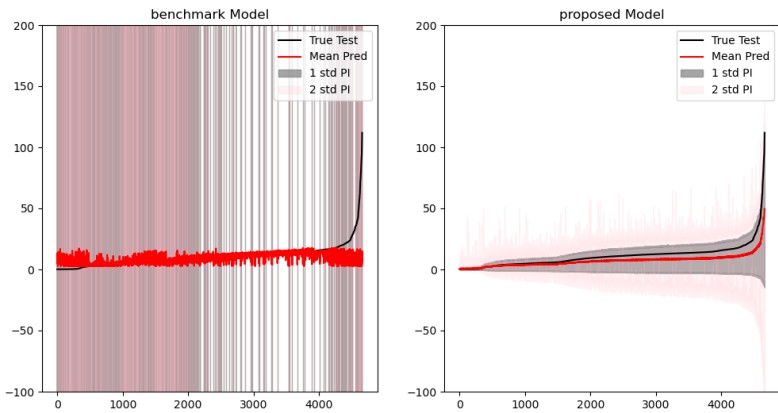

Figure 5: Predicted UQ for Benchmark Regression (left) vs proposed model(right). Again we see that the updated model is much better able to capture the UQ. With Uncertainty increasing after recovery rate increases beyond 40, which is a less dense region and has much fewer observations 4

prior work on uncertainty estimation [10] [6] and the utilization of neural networks for modeling probability distributions. In another line of work, Bayesian deep learning utilizes priors on network weights estimated with variational inference [1]. There also exists alternative techniques to estimate predictive variance like as dropout and ensembling which require sampling. [1]

# 5   Conclusion, limitations and future work

We propose an improvement over Deep Evidence Regression, specifically targeted to usecases where the target might follow weibull distribution. We then test the proposed method both on simulated and real world dataset in the context of Credit risk management. The proposed model exhibits enhanced suitability for applications in which the target variable originates from a weibull distribution, better capturing the uncertainty characteristics of such data. Although we have specifically tested the model in the credit risk domain, this method can be applicable to wide variety of safety critical regression tasks where the target variable follows a weibull distribution. The proposed approach thus serves as a valuable tool for capturing and quantifying uncertainty in cases characterized by weibull distributions, thereby enhancing the trustworthiness and explainability of model predictions, ultimately leading to improved confidence in the modeling process and the corresponding decision making.

However, we are not sure if the proposed approach would generalise to other distributions apart from Weibull. Additionally, the proposed model has only two outputs, which could limit its flexibility when compared to the benchmark model, which had four outputs from the neural network. In consequence the proposed model requires a deeper network architecture compared to the benchmark model. Furthermore we find that both the models exhibit a high sensitivity to regularization cost, which means that changes in the regularization coefficient can significantly impact the model's performance. The cylical learning rate as outlined in the [15], which proposes varying the learning rate between reasonable boundary values might be of help to mitigate this issue. Overall, these points suggest that both models have their strengths and weaknesses, and selecting the most appropriate model depends on the specific task requirements and considerations.

Considering the wide application of beta distributions in the credit risk domain, there may also be value in further extending the proposed technique to target variables characterized by beta distributions, as it has the potential to provide valuable insights and improved modeling capabilities in the context of credit risk management.

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
