# OpenReview forum: "Deep Evidence Regression for Weibull targets"
_NeurIPS.cc/2023/Conference — Submitted to NeurIPS 2023_

### Official Review · Reviewer_gHV7 · 2023-06-26

**Soundness:** 3 good
**Presentation:** 2 fair
**Contribution:** 2 fair
**Rating:** 4
**Confidence:** 3

**Summary:**

This paper aims to explore the application of a scalable UQ-aware deep learning technique, Deep Evidence Regression, and applies it to predict Loss Given Default. It extends the Deep Evidence Regression methodology to learn target variables generated by a Weibull process and provides the relevant learning framework. By testing on both simulated and real-world datasets in the context of credit risk management, the proposed method exhibits enhanced suitability for applications in which the target variable originates from a Weibull distribution, better capturing the uncertainty characteristics of such data.

**Strengths:**

1.This paper innovatively extends the Deep Evidence Regression methodology to learn target variables generated by a Weibull process and provides the relevant learning framework.
2.The article provides a clear and coherent description of the proposed method, including a thorough derivation of the relevant formulas.

**Weaknesses:**

1.The article contains several mathematical formula writing errors, such as the missing "dθ" in line 51 and Equation (16), the missing "-1" in Equation (10), and the inconsistencies in Equation (31) and Equation (33) with their original definitions.
2.The figure legends in the article do not indicate which parts of the content reference them or provide explanations for their content.
3.The experimental section lacks clear exposition, such as the specific settings of parameters k and λ for the Weibull distribution, as well as the target or subject for the MSE metric.

**Questions:**

1.Can the Weibull distribution still be selected if the LGD values are symmetric?
2.What is the purpose of calculating E[Z] in Section 2.3.1 and Var(Z) in Section 2.3.2? How are the calculated E[Z] and Var(Z) specifically used in the experiment?
3.How are the parameters k and λ of the Weibull distribution set in the experiment?
4.What is the specific object of MSE metric in the experiment?
5.Why is it claimed that the MSE of the benchmark and the proposed method in Table 1 are similar?

**Limitations:**

The proposed approach serves as a valuable tool for capturing and quantifying uncertainty in cases characterized by Weibull distributions. Therefore, the key to using this method lies in determining whether the data is suitable for the Weibull distribution. Furthermore, the applicability of the proposed method to distributions other than the Weibull distribution requires further investigation.

---

### Official Review · Reviewer_v9ki · 2023-07-04

**Soundness:** 3 good
**Presentation:** 2 fair
**Contribution:** 1 poor
**Rating:** 3
**Confidence:** 3

**Summary:**

Authors tackle the problem of uncertainty quantification for predicting credit risks. Concretely, they have applied a scalable UQ-aware deep learning technique, Deep Evidence Regression to predicting Loss Given Default with uncertainty. Authors argue that the conventional methods use for uncertainty quantification are too computationally and memory intensive and therefore they adopt Deep Evidence Regression as their statistical model.


They extend the framework of Deep Evidence Regression to predict targets generated with a Weibull distribution. The original method relies on the normally distributed targets which is not fitting for credit risk applications. Authors therefore re-derive the necessary formulas based on Weibull distributed targets; concretely they focus on log likelihood needed for training and the mean/uncertainty needed for predictions. Beyond that, authors adapt the regularizers found in the original paper to their purpose.

The new method is tested empirically on a synthetic dataset with points sampled from a Weibull distributions as well as a single real-world dataset focused on peer to peer mortgage lending data with recovery rate has been used as a proxy for Loss Given Default. Both datasets were used to test the Weibull-based approach against the original method relying on Gaussian distributed targets. The results show the modification made by the authors indeed helps with lower MSE and NLL on both test and train datasets.


**Strengths:**

1. Authors tackle an important problem of uncertainty quantification for risk assesment where robust, and efficient measures of uncertainty are crucial to ensure fair treatment of customers.
2. Authors provide an expansion of a well established method to make it much more attractive to specialized domains such as risk assesment. This work can be directly useful for practicioners in the field of risk assesment and indirectly useful for researchers in other fields who can adapt the Deep Evidential Regression to work with different, field-specific targets distributions.

**Weaknesses:**

1. This paper provides an incremental improvement over the original Deep Evidential Regression (DER) work. The derivation of DER with Webull distribution instead of original gaussian target distribution is interesting but the majority of what makes the method impactful remains unchanged.
2. Authors provide very limited empirical evaluation of their work. The single study with synthetic data provides a good proof of concept but gives little assurance of real-world impact of the work. The single real-dataset study is relatively limited and suffers from some issues: 1/ Authors do not have any baselines beyond the original methods while other methods, for example based on conformal predictions or quantile regression, could be competitive. Any other approach for uncertainty quantification would be useful to gauge the difficulty of the task. 2/ Authors only use a single dataset making the evaluation process less robust, it is possible that this method works well only for this particular dataset rather than for a generic class of problems. 3/ Authors only consider NLL and MSE as metrics for their evaluation, there are may more metrics to quantify uncertainty quantification (such as coverage) that could be used to make the evaluation more robust.
3. The presentation of the initial method is vague, I undrstand that the DER work is presented in its own paper but presenting it in more detail would make this paper more self-contained, especially given the reliance on the original work.

**Questions:**

1. Authors suggest that “the proposed model has only two outputs, which could limit its flexibility when compared to the benchmark model, which had four outputs from the neural network” which is a little unclear to me. I understand that the model (NN) is as flexible as many free parameters (weights) it has rather than outputs. Maybe this refers to the parametrization of the distribution (gaussian with 4 parameters and Weibull with 2) but in any case, I would like to get some clarity on this claim.
2. The model has a few hyperparameters such as c (regularization constant), it is currently unclear how were they chosen and what HPO procedure was used.

**Limitations:**

Authors provide a formulation for extending Deep Evidential Regression for problems where targets are Weibull distributed. However, they do not cover the extension of this method to problems with targets sampled from a different distribution. There is only a small class of problems where Weibull is appropriate and authors do not provide evidence that this approach generalizes beyond Loss Given Default estimation. I would appreciate additional real-world experiments thta could show whether this parametrization (choice of target distribution) can work beyond the single dataset chosen in the paper.

---

### Official Review · Reviewer_aVLB · 2023-07-04

**Soundness:** 2 fair
**Presentation:** 1 poor
**Contribution:** 2 fair
**Rating:** 3
**Confidence:** 4

**Summary:**

This paper introduces the utilization and extension of deep evidential regression for uncertainty estimation in credit risk prediction. The approach assumes a Weibull distribution for the target variable (e.g., LGD or a synthetic target). The authors modify the evidential regression mechanism to accommodate targets from this distribution, providing equations to illustrate the training process. Simple experiments are conducted on synthetic and real-world peer-to-peer lending datasets, showcasing potential improvements over vanilla deep evidential regression for credit risk management.

**Strengths:**

The incorporation and extension of evidential regression-based UQ into finance-related problems is the main contribution and the key strength of this paper.

**Weaknesses:**

Refer to Questions.

**Questions:**

1. What is the motivation for choosing the Weibull distribution? Can the authors comment on whether the method can be utilized for other off-the-shelf finance datasets where the target is distributed otherwise or has a high degree of heteroskedasticity ?

2. The experimental setup is very limited.  For the synthetic example, the parabolic function is essentially corrupted with noise drawn from a known Weibull distribution.  Therefore the performance of the proposed approach is expected to be better than the baseline.  Can the authors explain why the MSE of the baselines is better than the proposed approach ? It seems counterintuitve. How exactly are the metrics evaluated ?

3. No references to any figure has been made in the paper.  The figure captions are also very generic. The figures as well as the captions need to be enhanced.

4. How does this approach fare against other uncertainty estimation based methods for e.g, Monte-Carlo dropout ? I believe it is straightforward to implement and verify.

5. Does Figure 3 depict the epistemic uncertainty or the total uncertainties around every sample?

6. Eqn 3 and 4 are general descriptions qualifying Eqn 2.  It would be good if they are not written as equations but written as text.  This is the case with many other redundant equations mentioned in the paper.

7. Identified typos:

a) Line 9 - approach to - approach on

b) Line 23 - The citations [23], [24] can be provided at the end of the preceding sentence.

c) Line 35 - predicts the types - predicts the type

d) Line 52 - trained t --> trained to

e) Line 56 -  the fullstop can be removed before the citation [7]

f) Line 72 - real world datasets

g) Line 76 - rate parameter $k$

h) Line 91 - spelling of ensembling

i) Eqn 31,  the mean term E[\lambda] must be squared.

j) Line 107 - increases --> increase

k) Line 183 - Spelling of evidential needs to be correct

8. Discrepancies between title in the paper and the title shown in the submission page ? Why ?

**Limitations:**

The paper's experimental setup is quite limited and requires further motivation. The demonstrations illustrating how uncertainty quantification can benefit credit risk problems lack sufficient substantiation. Moreover, the paper has a limited exploration of related work, and the quality of references needs improvement. In its current stage, the paper requires significant improvement in terms of better problem motivation and comprehensive qualitative and quantitative evaluations prior to publication.

---

### Decision · Program_Chairs · 2023-09-21

**Decision:**

Reject

**Comment:**

While the paper addresses an important problem of uncertainty quantification and is well motivated by a real world use case in credit risk, the reviewers unfortunately reached a clear consensus that the paper was not yet ready for acceptance to NeurIPS in its current form.  Some of these issues are due to presentation and clarity, and are noted in some depth by reviewers in a way that will hopefully be useful to authors in revision.  However, beyond the presentation, there are also key questions unresolved about the approach itself, most notably around why the Weibull distribution is the most appropriate out of all the possible distributions that might be selected.  I do note that the authors did not engage in a rebuttal, which might have been helpful here.  Also, because we had three reviewers respond (despite recruiting additional reviewers), I have also read the paper carefully myself and agree with the clear reviewer consensus and key points.  I hope that the reviewer comments will be helpful to the authors in helping to strengthen or revise the work from here.